# DEEP CLASS CONDITIONAL GAUSSIANS FOR CONTINUAL LEARNING

## ABSTRACT

The current state of the art for continual learning with frozen, pre-trained embedding networks are simple probabilistic models defined over the embedding space, for example class conditional Gaussians. However, as of yet, in the task-incremental online setting, it has been an open question how to extend these methods to when the embedding function has to be learned from scratch. In this paper, we propose an empirical Bayesian framework that works by storing a fixed number of examples in memory which are used to calculate the posterior of the probabilistic model and a conditional marginal likelihood term used to fit the embedding function. The learning of the embedding function can be interpreted as using a variant of experience replay, which is a highly performative method for continual learning. As part of our framework, we decide which examples to store by selecting the subset that minimises the KL divergence between the true posterior and the posterior induced by the subset, which is shown to be necessary to achieve good performance. We demonstrate the performance of our method on a range of task-incremental online settings, including those with overlapping tasks which thus far have been under-explored. Our method outperforms all other methods, including several other replay-based methods, evidencing the potential of our approach.

## 1 INTRODUCTION

Real world use of deep learning methods can often necessitate dynamic updating of solutions on non-stationary data streams (Farquhar & Gal, 2018; Antoniou et al., 2020). This is one of the main problems studied in continual learning and as a result, continual learning has become of increasing interest to the machine learning community, with many proposed approaches (Parisi et al., 2019) and settings (Hsu et al., 2018; Antoniou et al., 2020; Delange et al., 2021). Currently, the two biggest challenges in continual learning are *catastrophic forgetting* and *positive transfer*. *Catastrophic forgetting* describes the common occurrence in learning where unconstrained deep models easily forget information derived from previous data after updating on other data. *Positive transfer* is the ability of a model, given the current data, to improve its understanding of previous data and of what future data might imply. While there have been significant steps taken to solve these problems (Delange et al., 2021; Mai et al., 2021), in many settings there are still gains to be made (Farquhar & Gal, 2018).

In common with many works in continual learning, the setting considered here is *task-incremental online learning* where a data stream is split into a sequential set of *tasks* and methods are given information about what the current task is (van de Ven & Tolias, 2019; Prabhu et al., 2020). Each task is encapsulated by a representative dataset (considered to be sampled i.i.d. from a task distribution) which is given to a method batch by batch. Different tasks will generally be associated with different distributions, as well as different target problems. The target problems are summarised in a task objective function. In our case the task objective is classification (Hsu et al., 2018), where the classes being considered vary between tasks. The overall objective of a method, after seeing all of the tasks, is to perform well on all of them, given constraints on the amount of memory used by the method.

Currently one of the best ways to approach continual learning is to use a frozen pretrained embedding function and define a simple probabilistic model on top to classify the data (Ostapenko et al., 2022;

Hayes & Kanan, 2020). However, in some real-world settings it is necessary to learn the embedding function online, for example it might be impossible to pretrain the embedding function due to a lack of data or due to distribution shifts a frozen pretrained encoder may become outdated and so perform badly (Ostapenko et al., 2022). Therefore, it is an interesting question to see how these methods can be adapted to settings when the embedding function must be continually learnt. We explore this question, reformulating it into an empirical Bayesian framework and propose a general approach: learning the parameters of the probabilistic model in a Bayesian manner while giving a method to learn the embedding function using a conditional marginal log-likelihood loss. The approach also stores a small number of previous examples, used to calculate the posterior of the probabilistic model and the conditional marginal log-likelihood loss. The learning of the embedding function can be seen as a variant of experience replay, this is beneficial as experience replay is one of the best performing continual learning methods (van de Ven & Tolias, 2019; Wu et al., 2022; Mirzadeh et al., 2020). Another key part of the approach is a method to select examples to store in memory, which is achieved by minimising the KL-divergence between the true posterior and the one induced by the subset of data to be stored in memory, which is shown to be necessary for the method to achieve good performance (see Section 6.3).

We also present a specific instantiation of our general approach, DeepCCG, where we use a class conditional Gaussian model with unknown mean and fixed variance on top of the neural network embedding function. We chose to explore this particular instantiation because its posterior is easy to compute and the class conditional Gaussian model has been shown to have state-of-the-art performance when using a frozen pretrained embedding function (Ostapenko et al., 2022; Hayes & Kanan, 2020). Additionally, in this case, our method for selecting what samples to store in memory reduces to selecting examples that preserve the means of the per-class clusters formed by the embedded data.

In our experiments we look at two specific settings, the commonly used disjoint tasks setting, where each task contains different classes to any other (Delange et al., 2021) and an underexplored shifting window setting, where there is an overlap between what classes are in each task. The reason we look at a setting with an overlap between tasks is to explore a methods ability for positive transfer, which is more rewarded in this setting as there is greater shared information (Bang et al., 2021). The results of our experiments show that DeepCCG performed the best out of all methods tested, gaining an average performance improvement of $2.145\%$ in terms of mean average accuracy, showing the potential of our approach.

## 2 RELATED WORK

When using frozen pretrained embedding functions in continual learning, simple metric-based probabilistic models have been shown to have state-of-the-art performance (Ostapenko et al., 2022; Hayes & Kanan, 2020; Pelosin, 2022). For example, Hayes & Kanan (2020) show that online linear discriminant analysis (LDA) on top of a frozen pretrained backbone is the best performing method in their experiments and Ostapenko et al. (2022) show that class-conditional Gaussian models perform the best in certain settings. These probabilistic models have the advantage that it is possible to learn the same estimate of parameters given any ordering of the data, including i.i.d orderings. However, in many cases frozen pretrained embedding functions cannot be used, which is the case we look at in this work, due to a lack of relevant data for pretraining or due to distribution shift a pretrained embedding function will perform badly (Ostapenko et al., 2022).

When the embedding function has to be learnt, there are three main paradigms for solving continual learning problems: *regularisation*, *parameter-isolation* and *replay* (Delange et al., 2021). *Regularisation* methods aim to prevent catastrophic forgetting by adding terms to the loss which try to prevent the network from erasing the information of previous tasks (Kirkpatrick et al., 2017; Huszár, 2018). *Parameter-Isolation* methods look at controlling what parameters of a neural network are used for what tasks (Mallya & Lazebnik, 2018), which can often be seen as a *hard* version of regularisation methods. Finally, *Replay* methods aim to solve continual learning problems by storing a subset of previously seen samples, which are then trained on alongside new incoming data. This is the approach our method is part of. Replay methods have been shown to have competitive if not the best performance across many settings (van de Ven & Tolias, 2019; Wu et al., 2022; Mirzadeh et al., 2020). The standard replay method is experience replay (ER) (Chaudhry et al., 2019b; Aljundi et al., 2019a), which in the setting explored in this work where we can not store all the current task's data,

selects samples to store using reservoir sampling (Vitter, 1985), we call this variant ER-reservoir. One of the main questions to be answered by a replay-based approach is how to select what samples to store in memory. While reservoir sampling has been shown to be very effective (Wiewel & Yang, 2021) there have been other methods proposed for sample selection, for example ones which use information-theoretic criteria (Wiewel & Yang, 2021) and others maximising the diversity of the gradients of stored examples (Aljundi et al., 2019b). There has also been a Bayesian method proposed to select samples called InfoGS (Sun et al., 2022), which is somewhat similar to our method but only uses the probability model to select samples, not for prediction or training, and unlike our work looks at *class-incremental learning*, where methods do not have access to task identifiers.

There has been considerable work on using Bayesian methods in continual learning (Nguyen et al., 2018; Ebrahimi et al., 2020; Kurle et al., 2020), perhaps inspired by the fact that true online Bayesian inference cannot suffer from catastrophic forgetting (Nguyen et al., 2018). Bayesian perspectives have mainly been used for regularisation based methods, where a popular approach is to use variational inference (Nguyen et al., 2018; Farquhar & Gal, 2019; Zeno et al., 2018). These variational inference methods focus on the offline continual learning setting where a learner has access to all of the data of a task at the same time and, in previous work, are often limited to being used in conjunction with small neural networks, mainly due to the need to sample multiple networks when calculating the loss (Henning et al., 2021; Nguyen et al., 2018). Therefore, these methods are not suited to the settings we consider in this paper. Bayesian methods have also been used in generative replay based approaches, where instead of storing and replaying real samples they use generated pseudo-samples (Rao et al., 2019). However, when it comes to replay, with real examples, there has been relatively little work on using Bayesian methods, specifically for empirical Bayesian methods where there is a requirement to prevent forgetting when fitting the embedding function, which we aim to help to fill in by proposing our method.

In this work we look at two types of settings, one where the sets of class labels are disjoint between tasks and another where there is an overlap been the classes seen in each task. Disjoint tasks is a commonly explored setting (Delange et al., 2021) while having some class overlap between tasks is not often studied, but arguably is more realistic. Slightly tangential to the settings considered here, there has been some work looking at class overlap settings where blurry tasks are considered (Prabhu et al., 2020; Bang et al., 2021; Aljundi et al., 2019b), that is when a disjoint task setting is modified by adding to each task a small number of samples sampled from any class, often around $10\%$ of the original task's size.

## 3   TASK-INCREMENTAL ONLINE CONTINUAL LEARNING

Task-incremental online continual learning is a setting where the learner is given a sequence of batches of data one by one, along with a task identifier for each batch. Different batches may be drawn from different tasks according to some unknown non-stationary process. The goal is for the learner to be able to perform well on all tasks at the end of training (Hsu et al., 2018; Prabhu et al., 2020).

Specifically, we consider classification tasks. Let $X$ denote a shared data space, and $C$ denotes the set of all classes being considered. Let $t \in T$ denote a particular task, which generates data $S_t = \{(x_i, y_i)|i = 1, 2, \ldots\}$, containing data instances $x_i \in X$ and labels $y_i \in C_t$, where $C_t \subset C$ are the subset of classes relevant for this task. For simplicity, we follow previous work (Delange et al., 2021) and assume the number of classes in each task is the same, i.e. $|C_t| = k, \forall t$.

The learner is provided with a temporal sequence (of length $N$) of data batches and task identifiers $((D_j, t_{D_j})|j = 1, 2, \ldots, N)$, where $j$ indexes batches and $t_{D_j}$ denotes the task identifier for batch $D_j$. At each time step $j$, the batch $D_j$ is a sample from a single task and the *task identifier* $t_{D_j}$ determines this task, indicating the classes $C_{t_{D_j}}$ which are relevant for that task. We use the notation $D_{<j}$ to denote $\{D_1, D_2, \ldots, D_{j-1}\}$, the data from all batches seen prior to the time step $j$.

During training, the learner receives each batch in sequential order; after training on a batch the learner must discard all the data it does not store in its memory buffer, which is of a fixed size (10 samples per-class in our experiments). At test time, the learnt model is given a previously-unseen test set from all the classes seen during training. This setting is commonly used in the continual learning literature (Chaudhry et al., 2019a; Prabhu et al., 2020).

It is often assumed that the set of classes of each task are disjoint (Bang et al., 2021; Prabhu et al., 2020); in our experiments we look at both the disjoint setting and an under-explored shifting window setting. We define the shifting window setting as follows. Let $C = \{c_i\}_{i=0}^k$ and construct task $t \in [0, \ldots, k - l + 1]$ by creating a dataset $S_t$ which consists of samples which belong to classes $c_t, \ldots, c_{t+l}$, where $l$ is the window length and where each sample can only belong to one task. In other words, a task $t$ contains samples from all the classes seen in the previous task except for the the class in the previous task with the smallest index and contains samples from a new class $c_{t+l}$. The reason we use a windowing scheme in one of our settings is to test a method's performance when there is an increased overlap between tasks and where similar tasks are located temporally near to each other, which is a common real world property. We want to explore having a large overlap between tasks so to allow the methods to more fully see what across-task information is useful to remember and therefore hopefully reward methods which allow for better forward and backward transfer. This is unlike the disjoint setting, where tasks can have little overlap and so discourages the use of methods which try to allow for a large amount of backwards transfer.

## 4 CONTINUAL LEARNING OF EMBEDDING FUNCTIONS AND PROBABILISTIC MODELS

To solve the problem of exploiting simple probabilistic models, like LDA, when the embedding function has to be learnt online, we propose an empirical Bayesian method (shown in Algorithm 1). Let $Z$ denote an embedding space. We assume that we have some embedding function $f_\phi : X \to Z$ and a fixed memory of size $m$, which counts the number of items (each an element of $X \times C \times T$) that can fit in a memory buffer. The memory buffer stores as one element a sample $(x, y)$ and its associated task index $t_{(x,y)}$—in common with previous work (Chaudhry et al., 2019a). We propose to use a Bayesian probabilistic model with parameters $\theta$, such that

$$p(y|x, t_{(x,y)}) = \int p(y|f_\phi(x), \theta, t_{(x,y)})p(\theta)d\theta, \tag{1}$$

and assume the labels are conditionally independent given the task, $\theta$ and the data instances (each an element of $X$). Therefore if $f_\phi$ is fixed, we can update our beliefs about the parameters on seeing batch $D_j$ and its task identifier $t_{D_j}$ by simply using Bayes rule, calculating the posterior $p(\theta|D_j, D_{<j})$. However, in our setting, we need to update $f_\phi$ continually, which would change the value of $p(\theta|D_j, D_{<j})$. We cannot recompute $p(\theta|D_j, D_{<j})$ as we are unable to store all the previous samples in memory. Therefore, we propose to store in the buffer a set of representative samples with which we can compute an approximate posterior after a change in $f_\phi$. To achieve this goal, after every batch, we select a subset of data points, from the previous memory and current batch, to store in memory. We choose the subset which minimises the KL divergence between the posterior distribution induced by the subset and the posterior distribution induced by the whole data,

$$M_{j+1} = \underset{M' \subseteq (D_j \times \{t_{D_j}\}) \cup M_j, |M'|=m}{\arg\min} \text{KL}(p(\theta|D_j, t_{D_j}, M_j)\|p(\theta|M')), \tag{2}$$

where $M_j$ is the memory which was carried forward when receiving the $j$th batch of data.

To update the embedding function, for each arriving batch $D_j$, task identifier $t_{D_j}$ and corresponding memory $M_j$, the method randomly subsamples $R_j \subset M_j$ of given size $r$. Then we update $f_\phi$ using

$$\phi_{j+1} = \phi_j + \eta\nabla_\phi \log p(D_j^Y, R_j^Y | D_j^Z, t_{D_j}, R_j^Z, S_{tasks}, M_j/R_j; \phi), \tag{3}$$

where $S_{tasks} = \{t_{(x,y)}|(x,y) \in R_j\}$, $\eta$ is a learning rate, $D_j^Z = \{f_\phi(x)|(x,y) \in D_j\}$, $D_j^Y = \{y|(x,y) \in D_j\}$ and $R_j^Z$ and $R_j^Y$ are defined likewise. This is a gradient step on a conditional marginal log-likelihood (i.e. we compute the marginal log-likelihood on some of the data having conditioned on the rest). Conditioning on part of the data helps to reduce the sensitivity of learning in respect to a potentially uninformative prior and has been shown in previous work to increase generalisation performance (Lotfi et al., 2022).[1]

---

[1]The marginal likelihood is the average likelihood over models generated from the prior, while the conditional likelihood is the average likelihood over models from the posterior induced by $M_j/R_j$. Averaging over an uninformative prior will often result in a flat marginal likelihood which is not informative of what embedding function fits the data well. Whereas, conditioning the marginal likelihood on $M_j/R_j$, the posterior we average over is more informative, having some belief of where the position of embedded data instances should be, and so provides a better signal to fit the embedding function.

---

**Algorithm 1** General approach, learning step at time $j$

---

1: **input** $D_j$ (current batch), $t_{D_j}$ (current task identifier), $M_j$ (current memory), $\phi_j$ (current embedding function)
2:
3: Update embedding function
4: $R_j = \text{UniformSample}(M_j)$
5: $\phi_{j+1} = \phi_j + \eta\nabla_\phi \log p(D_j^Y, R_j^Y | D_j^Z, t_{D_j}, R_j^Z, S_{tasks}, M_j/R_j; \phi)$
6:
7: Update memory
8: $M_{j+1} = \underset{M' \subseteq (D_j \times \{t_{D_j}\}) \cup M_j, |M'|=m}{\arg\min} \text{KL}(p(\theta|D_j, t_{D_j}, M_j)\|p(\theta|M'))$

---

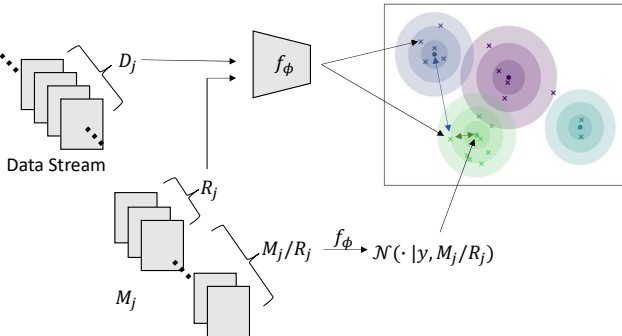

Figure 1: Diagram of DeepCCG's training routine. At time $j$ the learner is given a sample of data $D_j$ and has a memory set of stored datapoints $M_j$. The memory is split into the replay data $R_j$ and the rest $M_j/R_j$. Learning happens by taking a gradient step on the parameters of the embedding function $\phi$ using a conditional marginal likelihood function over $D_j$ and $R_j$, where $M_j/R_j$ is used to induce a posterior over the means of the per-class Gaussians and so define the conditional marginal likelihood function used. Therefore, training aims to move the data points into per-class clusters, by drawing the embedded examples of $D_j$ and $R_j$ towards their own class means and away from the other class means, as shown in the embedding space diagram in the figure.

One way to view of our update procedure for the embedding function $f_\phi$ is that it is a variant of experience replay, with a different loss function. Experience replay is a highly performing method to prevent forgetting (van de Ven & Tolias, 2019; Wu et al., 2022; Mirzadeh et al., 2020), so therefore our method to learn the embedding function should be resistant to forgetting. Additionally, we note that their exists methods to select what examples to replay (Aljundi et al., 2019a; Shim et al., 2021) which are complementary/orthogonal to this work, and could be combined with our approach. We further note that while our general formulation requires the continual learning setting to provide task identifiers, if $p(y|x, \theta, t_{(x,y)}) = p(y|x, \theta)$, where the distribution of $x$ can still shift between tasks, our general approach could be used in task-agnostic settings as well, which we see as a potential direction for future work.

## 5 DEEP CLASS CONDITIONAL GAUSSIANS

Having described the general method, we explore a specific instance of it in this paper, which we name DeepCCG. We use a neural network for the embedding function $f_\phi$. We define a class conditional Gaussian model in the embedding space $Z$ and use the inverse model of that class conditional Gaussian to define $p(y|z, t, \theta)$, where the parameters $\theta = \{\mu_y|y \in C\}$, that is the parameters of this model are the means of the per-class Gaussians. The reason we chose to use a class conditional Gaussian model is due to it having state-of-the art performance when using a frozen pretrained embedding function (Ostapenko et al., 2022; Hayes & Kanan, 2020) and that by using its associated conjugate prior, the posterior distribution for $\theta$ is tractable and easy to compute. Likewise, the marginal likelihood given by the inverse model is fully tractable. More formally, for each task $t$ we

define

$$p(y|z, t, \theta = \{\mu_c | c \in C\}) = \frac{p(z|y, \mu_y)p(y|t)}{\sum_{c \in C} p(z|c, \mu_c)p(c|t)}, \tag{4}$$

where

$$y|t \sim \text{Cat}(C_t, (1/|C_t|)\mathbf{1}) \tag{5}$$
$$z|y \sim \mathcal{N}(\mu_y, \mathbf{I}) \tag{6}$$
$$\mu_y \sim \mathcal{N}(\mathbf{0}, \mathbf{V}_0). \tag{7}$$

In this paper, we choose $\mathbf{V}_0 = a\mathbf{I}$, where in practice we take $a \to \infty$, and $\text{Cat}(C_t, (1/|C_t|)\mathbf{1})$ is the uniform categorical distribution over the classes of task $t$.

**Learning the embedding** To update the embedding function $f_\phi$, in response to the arrival of batch $D_j$, with task identifier $t_{D_j}$, and having received the memory buffer $M_j$, we take the following steps (shown in Figure 1). First, we select a set $R_j \subset M_j$, of size $r$, from the memory buffer. Then we condition on the rest of the examples, $M_j/R_j$, to obtain the posterior predictive distributions $p(y|f_\phi(x), t_{(x,y)}, M_j/R_j)$ for each sample $(x, y) \in D_j \cup R_j$, where $t_{(x,y)}$ is the task identifier for the sample. The posterior predictive distributions can be shown to be (see Appendix B for the derivation):

$$p(y|z = f_\phi(x), t_{(x,y)}, M_j/R_j) = \frac{\mathcal{N}\left(z | \overline{M_{j,y}^Z / R_j^Z}, (1 + \frac{1}{m-r})\mathbf{I}\right)}{\sum_{c \in C_{t_{(x,y)}}} \mathcal{N}\left(z | \overline{M_{j,c}^Z / R_j^Z}, (1 + \frac{1}{m-r})\mathbf{I}\right)} \tag{8}$$

where $M_{j,y}^Z = \{f_\phi(x) | (x, y') \in M_j \wedge y' = y\}$ are the embeddings of samples in the memory buffer with class $y$. $R_j^Z$ is defined likewise and we use the notation $\overline{S}$ to denote the mean of the elements of a set $S$. As in meta-learning (Garnelo et al., 2018; Hospedales et al., 2021), the posterior predictive distributions are used as a fixed likelihood term for each individual unseen data item. This leads to a log-marginal likelihood as the objective:

$$\phi_{j+1} = \phi_j + \eta \nabla_\phi \sum_{(x,y) \in D_j \cup R_j} \log(p(y|z = f_\phi(x), t_{(x,y)}, M_j/R_j)). \tag{9}$$

where $t_{(x,y)}$ is the task identifier associated with sample $(x, y)$.

**Sample selection** The second key component to DeepCCG is the selection of samples for the memory buffer. Because all the information available to inform the value of the parameters so far is encapsulated in the full posterior over the parameters, we target choosing a set of memory samples that best recreate that posterior, preventing as much parameter information loss as possible. Hence, to perform sample selection we minimise the KL divergence between two posterior distributions: the posterior over parameters induced by the new memory being optimized, and the posterior induced by the current batch and the old memory. We keep the number of samples in memory for each class balanced, therefore minimising the KL divergence is equivalent to minimising the per-class KL divergence which reduces to the squared Euclidean distance between the means of a class's embedded data. Formally,

$$M_{j+1,y} = \underset{M'_y \subseteq (D_{j,y} \times \{t_{D_j}\}) \cup M_{j,y}, |M'|=m}{\arg\min} (\text{KL}(p(\mu_y|D_{j,y}, M_{j,y})\|p(\mu_y|M'_y))) \tag{10}$$

$$= \underset{M'_y \subseteq (D_{j,y} \times \{t_{D_j}\}) \cup M_{j,y}, |M'|=m}{\arg\min} (\|\overline{D_{j,y}^Z \cup M_{j,y}^Z} - \overline{M_y'^Z}\|_2^2), \tag{11}$$

where $M_{j+1,y}$ is the new memory to be selected for class $y$, $M_{j,y}$ is the is the set of samples of class $y$ in memory when receiving the current batch of data and $D_{j,y}$ is the set of samples of class $y$ in the current batch.

Performing the minimization in Eq. 11 is computationally hard, so we utilise a relaxation of the problem using *lasso* (Hastie et al., 2009), whereby our method selects the new memory by assigning to each embedded input $z_i$ a zero-to-one weight $\beta_i$ and performing gradient decent on the loss

$$\mathcal{L}(\beta; D_{j,y}, M_{j,y}) = \|\overline{D_{j,y}^Z \cup M_{j,y}^Z} - \frac{1}{\|\beta\|_1} \sum_{\substack{i|(x_i, y_i) \in \\ D_{j,y} \cup M_{j,y}}} \beta_i z_i\|_2^2 + \lambda\|\beta\|_1. \tag{12}$$

Then, after termination, our method selects the $m$ samples with the largest weights to be the samples stored in memory, where they are stored along with their task identifiers.

# 6 EXPERIMENTS

## 6.1 EXPERIMENTAL SETUP

**Benchmarks** In our experiments we look at two different settings: disjoint tasks and a shifting window setting. Furthermore, in these settings we utilize three datasets CIFAR-100 (Krizhevsky, 2009), MiniImageNet (Vinyals et al., 2016) and CIFAR-10 (Krizhevsky, 2009), where both CIFAR-100 and MiniImageNet contain 100 classes, while CIFAR-10 contains 10 classes. For disjoint tasks, we split the datasets using the classes of the samples, splitting evenly the classes across a certain number of tasks and assigning all samples with that class to that task—this is often called the split tasks setting in previous work (Delange et al., 2021; Chaudhry et al., 2019a). We split CIFAR-10 into 5 tasks where there are 2 classes per task and for CIFAR-100 and MiniImageNet we split the dataset into 20 tasks with 5 classes per task. While in the shifting window setting we split the datasets up into tasks by fixing an ordering of the classes $c_1, \ldots, c_k$ and construct the $t$th task by selecting a set of samples from classes $c_t, \ldots, c_{t+l}$, where $l$ is the window length, no two task datasets contain the same example and each task has the same number of samples per-class. For CIFAR-10 we use a window length of 2 and for CIFAR-100 and MiniImageNet we use a window length of 5. Additionally, for all experiments we train with 500 samples per-class.

**Methods** We compare DeepCCG to representative methods of the main paradigms of continual learning. For regularisation methods we compare against a fixed memory variant of EWC (Huszár, 2018; Kirkpatrick et al., 2017) and for parameter-isolation methods we compare to PackNet (Mallya & Lazebnik, 2018). For replay methods, which includes DeepCCG, we compare against ER-Reservoir (Aljundi et al., 2019a), A-GEM (Chaudhry et al., 2019a), EntropySS (Wiewel & Yang, 2021), GSS (Aljundi et al., 2019b); where, EntropySS and GSS are memory sample selection strategies for ER. All methods are given task identifiers which state what classes are in a task and so what logits to mask for predicting samples of that task, which is identical to using a separate head per task in the disjoint task setting. We also compare against two baselines: SGD which is when learning is performed using SGD with no modification and Multi-Task which is a rough upper bound and is when we learn the base neural network offline, training on the same number of samples per batch as the replay methods tested, which is 20 (10 new samples and 10 replayed), with the same number of batches, to more fairly upper bound their performance. All methods use the same underlying embedding network for all experiments which is a modified ResNet18 with six times fewer filters across all layers and Instance Normalisation (Ulyanov et al., 2016) instead of Batch Normalisation layers (Ioffe & Szegedy, 2015), which is similar to other work (Mirzadeh et al., 2020; Farajtabar et al., 2019). Additionally, all methods use a batch size of 10, with all replay methods having the same sample replay size of 10 and a memory size of 10 samples per-class for all experiments.

**Metrics** We evaluate the methods using a standard metric for continual learning, average accuracy(Chaudhry et al., 2019a). The average accuracy of a method is the mean accuracy on a reserved set of test data across all tasks after training on all tasks. More formally, the average accuracy is $A = \frac{1}{T} \sum_{t \in T} a_t$, where $a_t$ is the accuracy of a method on the test data of task $t$ after training on all of the tasks.

## 6.2 RESULTS

In the shifting window setting, we see from Table 1 and Table 2 that DeepCCG performs the best out of all the methods compared. For CIFAR-100, MiniImageNet and CIFAR-10, DeepCCG gets mean average accuracies of $56.62\%$, $42.18\%$ and $74.13\%$, respectively, which is $2.57\%$, $1.14\%$ and $3.1\%$ better than any other method, respectively. ER-reservoir is the next best performing method for CIFAR-100 and MiniImageNet, showing that the memory selection strategies GSS and EntropySS hurt performance for this setting in these datasets. While for CIFAR-10, GSS outperforms ER-reservoir and achieves the second highest accuracy of $71.55\%$, indicating along with DeepCCG that for this dataset there are better ways of memory selection than uniformly sampling across the stream. It is important to note however that while DeepCCG performs better than the others it is still far

Table 1: Results of experiments on CIFAR-100 and MiniImageNet, where we report mean average accuracy with their standard errors across three independent runs.

| | CIFAR-100 | | MiniImageNet | |
|---|---|---|---|---|
| **Method** | **Shifting Window** | **Disjoint Tasks** | **Shifting Window** | **Disjoint Tasks** |
| EWC | $36.65 \pm 1.073$ | $42.39 \pm 0.787$ | $32.42 \pm 1.129$ | $29.57 \pm 0.689$ |
| PackNet | $40.21 \pm 0.961$ | $50.28 \pm 0.578$ | $34.15 \pm 1.280$ | $37.86 \pm 1.710$ |
| ER-reservoir | $54.05 \pm 0.626$ | $58.31 \pm 1.084$ | $41.04 \pm 1.539$ | $40.29 \pm 1.077$ |
| A-GEM | $29.01 \pm 1.449$ | $39.00 \pm 0.745$ | $26.97 \pm 1.264$ | $30.08 \pm 1.855$ |
| EntropySS | $51.80 \pm 0.700$ | $56.75 \pm 0.806$ | $40.03 \pm 0.609$ | $41.12 \pm 0.513$ |
| GSS | $48.20 \pm 0.332$ | $49.92 \pm 0.496$ | $37.91 \pm 0.493$ | $38.77 \pm 0.980$ |
| DeepCCG (ours) | $\mathbf{56.62 \pm 0.288}$ | $\mathbf{60.46 \pm 0.243}$ | $\mathbf{42.18 \pm 0.449}$ | $\mathbf{43.04 \pm 0.638}$ |
| SGD | $35.63 \pm 1.271$ | $42.50 \pm 1.403$ | $33.23 \pm 0.674$ | $31.61 \pm 0.696$ |
| Multi-Task (**UB**) | $90.42 \pm 1.866$ | $59.76 \pm 0.487$ | $89.81 \pm 0.250$ | $48.46 \pm 1.211$ |

Table 2: Results of experiments on CIFAR-10, where we report mean average accuracy with their standard errors across three independent runs.

| **Method** | **Shifting Window** | **Disjoint Tasks** |
|---|---|---|
| EWC | $58.68 \pm 1.883$ | $63.33 \pm 0.871$ |
| PackNet | $69.29 \pm 2.272$ | $66.97 \pm 1.471$ |
| ER-reservoir | $70.44 \pm 0.806$ | $66.71 \pm 0.896$ |
| A-GEM | $57.63 \pm 2.163$ | $57.28 \pm 2.606$ |
| EntropySS | $67.93 \pm 0.625$ | $64.96 \pm 0.911$ |
| GSS | $71.55 \pm 1.45$ | $67.30 \pm 1.268$ |
| DeepCCG (ours) | $\mathbf{74.65 \pm 2.00}$ | $\mathbf{69.29 \pm 0.889}$ |
| SGD | $63.21 \pm 2.088$ | $63.58 \pm 0.306$ |
| Multi-Task (**UB**) | $96.20 \pm 0.690$ | $73.37 \pm 1.820$ |

below the multi-task performance of $90.42\%$, $89.81\%$ and $96.20\%$ for CIFAR-100, MiniImageNet and CIFAR-10, respectively, unlike in the disjoint tasks setting. This shows that in the shifting window setting where there is more overlap between tasks and hence across-task knowledge transfer is more rewarded there is still much work to be done on continual learning methods.

For the disjoint tasks setting DeepCCG performs well, outperforming all the other methods tested. For CIFAR-100, DeepCCG achieves a mean average accuracy of $60.46\%$, which is $2.15\%$ higher than any other method; for MiniImageNet, our DeepCCG approach achieves a mean average accuracy of $43.04\%$, which is $1.92\%$ higher than any other method and for CIFAR-10, DeepCCG achieves a mean average accuracy of $69.29\%$ which is $1.99\%$ better than any other method (see Table 1 and 2). In this setting ER-reservoir is the second highest performing method for CIFAR-100 with GSS performing better than it and achieving the second highest performance for the CIFAR-10 dataset, obtaining a mean average accuracy of $67.30\%$ compared to ER-reservoir's $66.71\%$. As noted above, for the shifting window setting GSS also performed better than ER-reservoir for CIFAR-10 showing that for this dataset its memory selection method is effective. For MiniImageNet the second highest achieving method was EntropySS, which got a mean average accuracy of $41.12\%$, which indicates along with its performance for the shifting window setting on this dataset, that its memory selection method is better than GSS's for MiniImageNet.

## 6.3 ABLATION STUDY

One way to view DeepCCG is as a relative of ER-reservoir, where the methods differ in the sample section mechanism used and the probabilistic model defined on top of the embedding function. Therefore, to analyse where the performance of DeepCCG comes from, we ablate our method by separately changing these two components to what they are in ER-reservoir. The results are pre-

Table 3: Results of performing an ablation on DeepCCG, where we report results on Deep-CCG using reservoir sampling instead of its own sample selection method (DeepCCG-reservoir), when we replace the class-conditional Gaussian model with a standard classifier head (DeepCCG-standardHead) and when we do both modifications, in which case our method reduces to the ER-reservoir method. We report the mean average accuracy with their standard errors across three independent runs on the CIFAR-100 shifting window setting.

| Method | Average accuracy |
|---|---|
| ER-reservoir | $54.05 \pm 0.626$ |
| DeepCCG-reservoir | $49.95 \pm 0.743$ |
| DeepCCG-standardHead | $44.05 \pm 0.264$ |
| DeepCCG | $56.62 \pm 0.288$ |

sented in Table 3 and show that both components on their own do not improve performance over ER-reservoir but together they do. We go into detail of the ablation of both components, the sample selection mechanism and probabilistic model, in the next two paragraphs. Additionally, we perform an ablation on the memory size used for the replay methods tested in Appendix D, where we show that DeepCCG performs the best over all the memory sizes tested on.

**Sample selection** To explore the performance of our memory selection mechanism, we compare our full method to when we select samples using reservoir sampling (DeepCCG-reservoir), showing that the full method performs the best in the setting tested on. DeepCCG achieves a mean average accuracy $56.62\%$ whereas DeepCCG-reservoir obtains a lower mean average accuracy of $49.95\%$ (see Table 3). This shows that our memory selection mechanism is necessary for DeepCCG to obtain good performance. Additionally, we predict that our memory selection mechanism will also perform well in settings with noisy, OOD and/or imbalanced data, unlike reservoir sampling (Sun et al., 2022; Yoon et al., 2022; Aljundi et al., 2019b), which we leave to future work to validate.

**Class conditional Gaussian model** To understand the performance of using a class conditional Gaussian model instead of the standard discriminative softmax model, we look at the performance of DeepCCG without using the class conditional Gaussian model. That is, we evaluate the performance of DeepCCG when we replace the class conditional Gaussian model with a standard output head (a fully connected layer, then a softmax), while still using our sample selection method—calling it DeepCCG-standardHead. The results show that DeepCCG-standardHead performs poorly, achieving an average accuracy of $44.05\%$ (see Table 3), which is significantly worse than when using reservoir sampling, as in ER-reservoir, which gets a mean average accuracy of $54.05\%$. This is to be expected as our sample selection strategy is designed with the assumption that our embedding function will cluster the data into per-class clusters which may not be true when using a standard multi-head output layer, while when using a class conditional Gaussian model the embedding function will be encouraged to create per-class clusters.

## 7 CONCLUSIONS

In this work we have demonstrated that using a replay based empirical Bayesian procedure for continual learning is a promising direction and can expand the use of simple probabilistic models from when the embedding function is frozen to when it is learnt online. We have proposed a general approach to learn online both the embedding function and the probabilistic model defined on top of it. As part of the method we proposed a way of selecting samples to store in memory by approximately selecting the best subset that minimises the KL divergence between the posterior induced by all the currently accessible data and the posterior induced by the subset. We also presented a specific instantiation of our approach using a class conditional Gaussian model, DeepCCG. The results show that DeepCCG performs well for task-incremental online learning, outperforming all other methods in the commonly used disjoint tasks setting (Hsu et al., 2018) as well as the underexplored shifting window setting, where there is an increased overlap between the tasks. For future work, potential directions would be to look at adapting this approach to class-incremental learning and in settings with imbalanced and/or noisy data (Sun et al., 2022; Yoon et al., 2022; Aljundi et al., 2019b).

## 8 REPRODUCIBILITY STATEMENT

To aid in the reproduction of our experiments, we provide a detailed explanation in Section 6.1 of the experimental setup and provide additional experimental details in Appendix C. Also, we provide the code used for the experiments in the supplementary material.

## 9 ETHICS STATEMENT

Continual Learning is a growing field and has the potential to have a large societal impact in the future. For example, a potential application of continual learning is for privacy-aware machine learning on mobile devices, where resource constraints mean that often a continual learning solution is necessary. Another example of the potential use of continual leaning is for the adaptation of large models, as re-training large models from scratch when new data becomes available is expensive and environmentally costly, therefore continual learning methods may provide a way of updating these large models with minimal resource costs. So, this means that research in continual learning could have a positive impact on society, however it is also important to realise that without conscious effort continual learning may have negative impacts as well. For instance, when deploying continual learning in the real world, mitigating social bias will be a problem and importantly solutions proposed when using static datasets—for example, modifying the dataset in some way—are often not straightforwardly applicable in continual learning. Therefore, before continual learning methods can be used fully in the real world many problems, known and potentially unknown, need to be addressed but if through future research they are addressed, continual learning can have a positive societal impact.

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

## A   ALGORITHM FOR DEEPCCG

---

**Algorithm 2** DeepCCG, learning step at time $j$

---

1: **input** $D_j$ (current batch), $t_{D_j}$ (current task identifier), $M_j$ (current memory), $\phi_j$ (current embedding function)
2:
3: Update embedding function
4:   $R_j = \text{UniformSample}(M_j)$
5:   $\phi_{j+1} \leftarrow \phi_j + \eta \nabla_\phi \sum_{(x,y) \in D_j \cup R} \log(p(y|z = f_\phi(x), t_{(x,y)}, M_j/R_j))$
6:
7: Update memory
8: **for** each class $c$ in $M_j$ **do**
9:     initialise $\beta$
10:    **for** 1 to $B$ **do**
11:       $\beta \leftarrow \beta + \eta \nabla_\beta \mathcal{L}(\beta; D_{j,c}, M_{j,c})$
12:    **end for**
13:    Set $M_{j+1}$ to be the set of samples, with their task identifiers, with the $m$ largest $\beta$ values
14: **end for**

---

## B   DETAILS OF LEARNING THE EMBEDDING FUNCTION

For our deep class conditional Gaussian method (DeepCCG), to compute the update to the embedding function $f_\phi$, when given a batch $D_j$ and a corresponding memory $M_j$, we proceed using the following steps. Firstly, we randomly sample $R_j \subset M_j$ of size $r$ from memory. Then using only the other samples in memory, $M_j/R_j$, we compute the posterior density for each class mean— $\mu_c$, with $c \in C$:

$$p(\mu_c|M_j/R_j) = p(\mu_c|M_{j,c}^z/R_j^z) \tag{13}$$

$$= \mathcal{N}\left(\mu_y|\overline{M_{j,c}^z/R_j^z}, \frac{1}{m-r}\mathbf{I}\right), \tag{14}$$

where $M_{j,c}^z = \{f_\phi(x)|(x,y) \in M_j \wedge y = c\}$ are the embeddings of points in the memory buffer with class $y$. $R_j^z$ is defined likewise and we use the notation $\overline{S}$ to denote the mean of the elements of a set $S$. Then we compute the posterior distribution of the embedded inputs $z \in D_j^z \cup R_j^z$ for each class $c \in C$ utilizing

$$p(z|c, M_j/R_j) = p(z|c, M_{j,c}^z/R_j^z) \tag{15}$$

$$= \int p(z|c, \mu_c)p(\mu_c|M_{j,c}^z/R_j^z)d\mu_c \tag{16}$$

$$= \mathcal{N}\left(z|\overline{M_{j,c}^z/R_j^z}, (1 + \frac{1}{m-r})\mathbf{I}\right). \tag{17}$$

Next, we compute the posterior predictive for each sample $(x, y) \in D_j \cup R_j$ with a task identifier $t_{(x,y)}$, which is known for samples in the current batch and is stored by our method for samples stored in memory, and where $z = f_\phi(x)$ using

$$p(y|z, t_{(x,y)}, M_j/R_j) = \frac{p(z|y, M_{j,y}^z/R_j^z)p(y|t_{(x,y)})}{\sum_{c \in C} p(z|Y = c, M_{j,c}^z/R_j^z)p(Y = c|t_{(x,y)})} \tag{18}$$

$$= \frac{p(z|y, M_{j,y}^z/R_j^z)}{\sum_{c \in C_{t_{(x,y)}}} p(z|Y = c, M_{j,c}^z/R_j^z)} \tag{19}$$

Finally, we update the embedding function by performing a gradient step using the formula

$$\phi_{j+1} = \phi_j + \eta \nabla_\phi \sum_{(x,y) \in D_j \cup R_j} \log(p(y|z = f_\phi(x), t_{(x,y)}, M_j/R_j)), \tag{20}$$

which can be seen as a per-sample conditional marginal log-likelihood.

By using Eq. 17 and 19, the closed form of the posterior predictive distribution for a sample $(x, y)$ with task identifier $t_{(x,y)}$ and where $z = f_\phi(x)$ is

$$p(y|z, t_{(x,y)}, M_j/R_j) = \frac{p(z|y, M_{j,y}^z/R_j^z)}{\sum_{c \in C_{t_{(x,y)}}} p(z|Y = c, M_{j,c}^z/R_j^z)} \tag{21}$$

$$= \frac{\mathcal{N}\left(z|\overline{M_{j,y}^z/R_j^z}, (1 + \frac{1}{m-r})\mathbf{I}\right)}{\sum_{c \in C_{t_{(x,y)}}} \mathcal{N}\left(z|\overline{M_{j,c}^z/R_j^z}, (1 + \frac{1}{m-r})\mathbf{I}\right)} \tag{22}$$

## C   ADDITIONAL EXPERIMENTAL DETAILS

In addition to the experimental details mentioned in the main body of the paper, there are a few more details to mention. Firstly, for methods with hyperparameters we performed a grid search using the same experimental set up as the real experiments and $10\%$ of the training data as validation data. This led to the selection of the hyperparameters of EWC and PackNet, the two methods with free hyperparameters, shown in Table 4. Secondly, we use the same learning rate of $0.1$ and the standard gradient decent optimiser for all methods. These were chosen by looking at the commonly selected values in previous work (Mirzadeh et al., 2020) and were shown to be performative for all methods tested. Finally, all experiments were run on a laptop with a single NVIDIA GeForce GTX 1050 GPU.

Table 4: Values selected for the hyperparameters of EWC and PackNet, which are the regularisation coefficient and the percentage of available filters to be used per task, respectively. SW stands for shifting window and DT stands for disjoint tasks.

| Method | CIFAR-100 SW | CIFAR-100 DT | MiniImageNet SW | MiniImageNet DT | CIFAR-10 SW | CIFAR-10 DT |
|---|---|---|---|---|---|---|
| EWC | 6 | 6 | 2 | 9 | 1 | 6 |
| PackNet | 0.1 | 0.1 | 0.05 | 0.2 | 0.3 | 0.3 |

## D   RESULTS ON USING DIFFERENT MEMORY SIZES

Table 5: Results of experiments looking at the effect of memory buffer size $m$ for the replay methods tested, using the shifting window setting on CIFAR-100. We report mean average accuracy with their standard errors across three independent runs.

| Method | m=750 | m=1000 | m=1250 |
|---|---|---|---|
| ER-reservoir | $53.17 \pm 0.656$ | $54.05 \pm 0.626$ | $55.22 \pm 0.592$ |
| A-GEM | $27.18 \pm 1.091$ | $29.01 \pm 1.449$ | $27.87 \pm 0.172$ |
| EntropySS | $50.52 \pm 0.725$ | $51.80 \pm 0.700$ | $53.34 \pm 0.372$ |
| GSS | $47.15 \pm 0.766$ | $48.20 \pm 0.332$ | $46.18 \pm 0.341$ |
| DeepCCG (ours) | $\mathbf{53.42 \pm 0.460}$ | $\mathbf{56.62 \pm 0.288}$ | $\mathbf{57.98 \pm 0.514}$ |

One additional useful experiment is looking at the relationship between performance and the size of memory used for DeepCCG. Therefore, in Table 5 we show the performance of DeepCCG and the other replay methods compared against with varying memory size. Table 5 shows that when the memory size is increased to $m = 1250$, DeepCCG has an improvement in average accuracy relative to other methods, as it achieves $2.76\%$ better than any other method for $m = 1250$, while for $m = 1000$ it achieves $2.57\%$ better than any other method, a $0.19\%$ improvement. When the memory size is decreased to $m = 750$, we see that DeepCCG's performance drops more than other methods as it is only $0.25\%$ better than other methods in this case. Therefore, our experiments show that compared to other replay methods DeepCCG's performance increases the most when $m$ increases and that for small buffer sizes DeepCCG performs less well, potentially due to the fact that

the examples in the memory buffer are used to infer the posterior over the means of the per-class Gaussians and so the method needs a given amount of examples to specify the means well. We also note that in our experiments the performance ranking of the methods does not change with $m$.

