# OpenReview forum: "Deep Class Conditional Gaussians for Continual Learning"
_ICLR.cc/2023/Conference — Submitted to ICLR 2023_

### Official Review · Reviewer_juyJ · 2022-10-23

**Confidence:** 4
**Correctness:** 3
**Technical Novelty And Significance:** 2
**Empirical Novelty And Significance:** 2
**Recommendation:** 5

**Clarity, Quality, Novelty And Reproducibility:**

* The paper is written clearly for the most part.
* The novelty of the paper is limited.

**Strength And Weaknesses:**

Strengths:
* A simple method that seems to work well compared to the baselines.
* The method was evaluated in two settings, disjoint tasks and a shifting window which shows that it is relatively robust.
* The paper is easy to follow and understand.
* Code was provided, which is noteworthy.

Weaknesses/Questions:
* Although I am not an expert in the field, it seems that the true novel part is the KL divergence loss for selecting the memory bank. I would imagine that there were other strategies for making this selection in the continual, life-long, or incremental learning setups, so why this strategy is better than others?
* Following the last question, Given that the authors chose a Gaussian class conditional distribution with unit covariance, I can imagine that the resulting selection strategy appeared in other papers as a frequentist loss function. Have the authors checked that?
* Lastly on the same subject, why did the authors assume a unit covariance? is it a realistic assumption? I would imagine that even an isotropic covariance matrix that requires inference on an additional parameter per class will be more suitable.
* I wonder what is the computational overhead of the two-step learning algorithm (i.e., learning the embedding and learning the probabilistic model on of them at each iteration), can you quantify that using wall-clock time or FLOPS?
* To me it seems that most baselines are outdated, how does the method compare to newer ones?
* It seems that the authors missed an important related work [1, 2] that share some of the assumptions of this work (for instance, [1] also assume that classes are grouped in a Gaussian ball in the feature space). In general, I wonder how this method or alternative methods that use GPs will work on the tasks presented in this paper. This seems like an important baseline.
* Which of the baselines learn the embedding function and which isn't? It seems like important information and I didn't find an answer to that.

[1] Achituve, I., Navon, A., Yemini, Y., Chechik, G., & Fetaya, E. (2021, July). GP-Tree: A Gaussian Process Classifier for Few-Shot Incremental Learning. In International Conference on Machine Learning (pp. 54-65). PMLR.

[2] Titsias, M. K., Schwarz, J., Matthews, A. G. D. G., Pascanu, R., & Teh, Y. W. (2019, September). Functional Regularisation for Continual Learning with Gaussian Processes. In International Conference on Learning Representations.

**Summary Of The Paper:**

The authors proposed an approach for learning embedding function and a probabilistic model on top of it for task-incremental online learning. The method has two update steps, first updating the model parameters based on the log-marginal likelihood of the probabilistic model and then update a memory bank of class representative examples by optimizing the kl divergence between a current posterior and a candidate posterior which considers data from the current task. The authors presented an instance of this algorithm with Gaussian distributions. The authors presented results of their method on CIFAR-10/100 and miniImageNet datasets in two settings, disjoint tasks (in terms of classes), and a shifting window.

**Summary Of The Review:**

I think that the paper is nice, but I am not sure that in terms of novelty, the approach, and the compared baselines it is good enough to be accepted. I am willing to change my mind in light of new information from the authors.

---

> ### Author Response · Authors · 2022-11-15
> **Author Response, Part 1**
>
> Dear reviewer, thank you for your considered questions and comments. We provide answers to your comments and questions below.
>
> ***1. Although I am not an expert in the field, it seems that the true novel part is the KL divergence loss for selecting the memory bank. I would imagine that there were other strategies for making this selection in the continual, life-long, or incremental learning setups, so why this strategy is better than others?***
>
> We compare to two other methods which propose sample selection strategies as opposed to using reservoir sampling, GSS and EntropySS. Our results show that our method performs better than these two methods. Additionally, in our ablations we show that without using our sample selection mechanism DeepCCG does not perform well at all, hence our sample selection mechanism is necessary for our method to be performative.
>
> ***2. Following the last question, Given that the authors chose a Gaussian class conditional distribution with unit covariance, I can imagine that the resulting selection strategy appeared in other papers as a frequentist loss function. Have the authors checked that?***
>
> Yes, we have looked extensively to see if our sample selection method has been used before in continual learning and to the best of our knowledge it has not. We do note though that there are ever more and more continual learning papers being produced so it is still possible for some paper to have slipped through the net.
>
> ***3. Lastly on the same subject, why did the authors assume a unit covariance? is it a realistic assumption? I would imagine that even an isotropic covariance matrix that requires inference on an additional parameter per class will be more suitable.***
>
> As part of this research project one of the approaches we tried was to learn the covariance matrix, however we found that it did not perform as well as having a fixed unit covariance matrix. Additionally, we note that a part of learning the nonlinear embedding function can be seen as fitting an implicit covariance matrix, so therefore learning covariance matrices do not add much flexibility as the embedding function already does most of the work associated with structured covariances.
>
> ***4. I wonder what is the computational overhead of the two-step learning algorithm (i.e., learning the embedding and learning the probabilistic model on of them at each iteration), can you quantify that using wall-clock time or FLOPS?***
>
> For our CIFAR10 experiments, running it on a GeForce RTX 2080 Ti, one run of DeepCCG took 38 seconds when a run of ER-reservoir took 36 seconds and the slowest method, GSS, took 84 seconds. In general, our method is more computationally expensive than ER-reservoir, our main comparison point, but is not the most expensive method to run in our experiments. We will elaborate this in the paper.
>
> ***5. To me it seems that most baselines are outdated, how does the method compare to newer ones?***
>
> To the best of our knowledge, for the online task-incremental continual learning setting ER-reservoir is highly competitive and is still often SOTA, which is why our paper has it as our main comparison. Additionally, as mentioned above, we compare to GSS, which is a well known method and is the best sample selection method we know of for online task-incremental settings. Lastly, we note that currently a lot of focus in continual learning is on offline, class-incremental and/or other types of settings, which is perhaps why there is less by way of more recent work that is consistently SOTA for the setting we consider.
>
> ***6. It seems that the authors missed an important related work [1, 2] that share some of the assumptions of this work (for instance, [1] also assume that classes are grouped in a Gaussian ball in the feature space). In general, I wonder how this method or alternative methods that use GPs will work on the tasks presented in this paper. This seems like an important baseline.***
>
> Thank you for pointing out these two works. They are interesting. However, they do not look at the same continual learning setting as ours and so it is not easy to see how they can be made directly comparable to our work. GP-Tree [1] looks at the Few-Shot Incremental Learning setting where the first task is very large and then subsequent tasks a few-shot only containing a few samples each, which is quite different to the setting we look at. FRCL [2] looks at a offline setting where you can store all of the current tasks data at the same time and also looks at when you need to infer task identifiers when learning, which is also quite different from our setting. Additionally, we do not currently know of work looking at GPs for the online task-incremental continual learning setting; it is something we considered.

---

> > ### Author Response · Authors · 2022-11-15
> > **Author Response, Part 2**
> >
> > ***7. Which of the baselines learn the embedding function and which isn't? It seems like important information and I didn't find an answer to that.***
> >
> > Sorry, we should of been more clear in the paper but all methods in the experiments learn the embedding function as in this paper we are interesting in the case where the embedding function has to be learnt.
> >
> > ***8. The novelty of the paper is limited.***
> >
> > We see the contributions of how to learn both the embedding function and class conditional Gaussian model together and the sample selection mechanism as novel, performative and of use to the continual learning community. This is because, to the best of our knowledge we are the first to propose how to probabilistically learn deep class conditional Gaussian models in a continual learning setting where the embedding function has to be learnt, which is a much tougher problem then when the embedding function is fixed. Also, while the conditional marginal log-likelihood has been used in previous work, to the best of our knowledge we are the first to use it in continual learning and has a special meaning in continual learning as it can be seen as a form of replay, which we show in the paper. Additionally, as mentioned in our answers to your other comments, we present a new sample selection method, which in our abations is shown to be necessary for DeepCCG to achieve good performance. Lastly, we also look at the shifting window setting which to the best of our knowledge has not been looked at before and show that currently continual learning methods perform badly on it compared to the multi-task upperbound, showing there is still problems to be solved in the online task-incremental setting. Therefore, we see our paper as both presenting a new method DeepCCG consisting of two novel components, a sample selection mechanism and a way to lean both the class conditional Gaussian model and embbeding function online, and a new unsolved setting.
> >
> > *** ***
> > We hope that we have sufficiently answered your questions, especially about our sample selection method and experiments, and that our answers are taken into account when reconsidering your review.
> >
> > [1] Achituve, I., Navon, A., Yemini, Y., Chechik, G., \& Fetaya, E. (2021, July). GP-Tree: A Gaussian Process Classifier for Few-Shot Incremental Learning. In International Conference on Machine Learning (pp. 54-65). PMLR.
> > [2] Titsias, M. K., Schwarz, J., Matthews, A. G. D. G., Pascanu, R., \& Teh, Y. W. (2019, September). Functional Regularisation for Continual Learning with Gaussian Processes. In International Conference on Learning Representations.
> > [3] An Investigation of Replay-based Approaches for Continual Learning , (IJCNN 2021) by Bagus, Benedikt and Gepperth, Alexander

---

> > > ### Comment · Reviewer_juyJ · 2022-11-17
> > > **Thank you for the response**
> > >
> > > I would like to thank the authors for the thorough response. I decided to raise the score to 5.
> > > I still believe that the novelty of the paper is limited and that the compared baseline methods are not sufficient. Therefore, I didn't raise the score further.

---

### Official Review · Reviewer_e2L5 · 2022-10-25

**Confidence:** 3
**Clarity, Quality, Novelty And Reproducibility:** The method is clear, novel, and repro…
**Correctness:** 4
**Technical Novelty And Significance:** 3
**Empirical Novelty And Significance:** Not applicable
**Recommendation:** 6

**Strength And Weaknesses:**

Strengths:
- The paper is well written and easy to follow
- The approach is relatively novel. The main focus in continual learning is on fixed backbone, also the shifted setting got relatively little attention.
- The results seem promising compared to other baselines.

Weaknesses:
- Not enough comparisons, cifar10 with only 10 classes is not a good benchmark for this method and cifar100 is a relatively easy benchmark. Would be better to compare on other benchmarks like CUB. Also use only a scaled-down version of resnet18
- Overall results aren't that good, maybe because the difficult setting but also could be due to the network used. Would be more convincing if it would work with a larger network

**Summary Of The Paper:**

The paper addresses the problem of lifelong learning, specifically without a fixed backbone by using a Bayesian approach. The approach shows good results in a challenging setting.

**Summary Of The Review:**

The paper addresses an important challenge and gets improved results compared to other approaches

---

> ### Author Response · Authors · 2022-11-15
> **Author Response**
>
> Dear reviewer, thank you for your review. We provide the answers to your comments below.
>
> ***1. Not enough comparisons, cifar10 with only 10 classes is not a good benchmark for this method and cifar100 is a relatively easy benchmark. Would be better to compare on other benchmarks like CUB. Also use only a scaled-down version of resnet18.***
>
> We chose our experimental setup by looking at and replicating the most used setups in previous work [1-4], where CIFAR10 and CIFAR100 are two of the main datasets used and it is very common to use scaled down ResNets. We emphasise that in the continual learning setting, both of these design decisions still present a significant challenge.
>
> ***2. Overall results aren't that good, maybe because the difficult setting but also could be due to the network used. Would be more convincing if it would work with a larger network.***
>
> To the best of our knowledge, the results are the best results yet published for this setting and experimental setup, so we are confused about the suggestion that results are not good. Large performance jumps are a challenge, especially in the disjoint task setting where performance of the multi-task upper-bound is close to what our and other methods get. So, while we agree the performance improvement of DeepCCG is not huge, it is still of note in the online task-incremental setting we consider in the paper.
>
> *** ***
> We hope that we have answered your questions sufficiently and that you consider them when you reconsider your review.
>
> [1] Online Continual Learning with Maximal Interfered Retrieval (Aljundi et al., 2019a) [2] Gradient Based Sample Selection for Online Continual Learning (Aljundi et al,. 2019b) [3] On Tiny Episodic Memories in Continual Learning (Chaudhry et al., 2019a) [4] Efficient Lifelong Learning with A-GEM (Chaudhry et al.,2019b)

---

> > ### Comment · Reviewer_e2L5 · 2022-12-12
> > **Datasets**
> >
> > Regarding datasets, three of the four papers you cited, [1,3,4], also use another, more challanging datasets such as CUB or miniImagenet so your claims of "replicating the most used setups in previous work" don't hold up with my claim of not enough comparisons.

---

> > > ### Author Response · Authors · 2022-12-12
> > > **Author response**
> > >
> > > Thank you for your comment, we do perform experiments on MiniImageNet as shown by Table 1.

---

> > > > ### Comment · Reviewer_e2L5 · 2022-12-12
> > > > **My mistake**
> > > >
> > > > I have no idea how I missed the miniImagenet results. Sorry for the mistake

---

> > > > > ### Author Response · Authors · 2022-12-12
> > > > > **Author Response**
> > > > >
> > > > > No worries, we all make mistakes.

---

### Official Review · Reviewer_rHsD · 2022-11-04

**Confidence:** 4
**Correctness:** 2
**Technical Novelty And Significance:** 3
**Empirical Novelty And Significance:** 2
**Recommendation:** 5

**Clarity, Quality, Novelty And Reproducibility:**

Clarity and Quality: The writing and presentation in the paper can be improved significantly. The section 4 and section 5 can be combined to clarify that the probabilistic model and the embedding network are updated separately.  There are also many grammatical errors that can be fixed.

Novelty: The novelty seems limited given that the probabilistic model is based on a simplified class-conditional Gaussian model (LDA) which was also used in Ostapenko et al. (2020). Moreover, the idea of using a subset of samples to determine the conditional marginal log-likelihood (Equation 3) was also explored in Lofti et al., (2022), although its application in continual learning seems new.



**Strength And Weaknesses:**

Strengths

- The proposed DeepCCG seems to do reasonably well in the task-incremental setting compared to some of the considered baselines.
- The experiment design of using a moving window to simulate task overlap has not been explored before.

Weakness

- *Missing related work*: There is an entire class of methods that do expansion-based [1-5] continual learning that have not been discussed in the related work. [1-4] also use a Bayesian framework (though for expansion) for continual learning. These methods also learn the entire model without using any pre-trained/frozen embedding network. This class of methods should be discussed in the related work.
- *Replay Baseline*: Replay based on generative models [5-6] has not been considered as a baseline.
- The writing and the presentation in the paper can be improved significantly. In terms of presentation, section 4 seems like an over-generalization of the methodology presented in section 5. Since the paper only discusses a simple variant of the general framework, which uses a class conditioned Gaussian to define the probabilistic model, section 4 seems a bit redundant.
- Some of the claims over-state the contribution, for e.g. Pg 5 section 4 (last para): "our general approach can be used in task-agnostic settings as well" -- The task-agnostic continual learning is a much more challenging setting than task incremental. Without any empirical evidence, this claim is not well supported.
- The sample selection strategy based on the KL divergence of the two posteriors in equation 11 assumes that the training data in each task is same across the tasks? From equation 11, it seems that given a new task with large number of samples, the new memory would be biased towards the new task, which could easily cause catastrophic forgetting of previous tasks. However, the paper only assumed settings with equal number of samples across all tasks.
- The paper only considers the task-incremental setting, which is known to be a relatively easier setup than other challenging continual learning settings, including class-incremental and task-free continual learning.


Other Minor concerns:
- The notation in the paper can be improved:
  - In equation 1, the marginalized likelihood is a function of the embedding network. Making this precise in the notation (e.g. $p_\phi (y|x,t_{x,y})$) would make the discussion that follows in section 4 clearer.
- In section 6.1 under benchmarks: "in shifting windows ... there is no overlap between any two of the task" -- Is this a typo? There is overlap with shifting windows, right?


[1] A Neural Dirichlet Process Mixture Model for Task-Free Continual Learning, (Lee et al., (2020))
[2] Continual Learning using a Bayesian Nonparametric Dictionary of Weight Factors (Mehta et al., (2021))
[3] Bayesian structure adaptation for continual learning, (Kumar et al. (2020))
[4] Hierarchical Indian Buffet Neural Networks for Bayesian Continual Learning, (Kessler et al, (2020))
[5] Efficient Feature Transformations for Discriminative and Generative Continual Learning, (Kumar et al., (2021))
[6] Deep Generative Replay, (Shin et al. (2017))


**Summary Of The Paper:**

The paper proposes using an empirical Bayesian framework to simultaneously learn an embedding model ($f_\phi$) and a probabilistic model (parameterized w/ $\theta$) for the continual learning setting. To that end, $f_\phi$ is updated using a variant of experience replay through a running memory of samples, where the memory samples are updated with each task such that the KL divergence between the true posterior for $\theta$ and the posterior with the new memory is minimized. While updating the embedding network, the authors use marginal log-likelihood wrt $y$ of a subset of data conditioned on the rest. The paper presents one instance of the proposed framework referred to as DeepCCG which uses conditional Gaussian model as the probabilistic model. Several experiments show the DeepCCG performs better than a few selected baselines in two different (disjoint and shifting window) task incremental settings.

**Summary Of The Review:**

Based on the above weaknesses and concerns, my recommendation is to reject the paper in its current state. While the idea of using DCCG w/ experience replay is interesting, there are some major concerns in terms of clarity and presentation. The baselines and related work can also be improved significantly.

---

> ### Author Response · Authors · 2022-11-15
> **Author Response, Part 1**
>
> Dear reviewer, we thank you for your comments and questions, which we think will improve the quality of our submission and provide answers to them here.
>
> ***1. Missing related work: There is an entire class of methods that do expansion-based [1-5] continual learning that have not been discussed in the related work. [1-4] also use a Bayesian framework (though for expansion) for continual learning. These methods also learn the entire model without using any pre-trained/frozen embedding network. This class of methods should be discussed in the related work.***
>
> You are quite right. We focus on fixed-memory methods in the paper and due to the differing assumptions made about resource constraints the setting we look at requires non-expansion based methods. Given the space and the vastness of the field, we focused background work on the areas directly related and did not elaborate on expansion-based methods. Also, for the same reasons, previous work on replay methods do not discuss and/or compare against expansion-based methods [7-10]. However, we do mention parameter-isolation methods in our related work which contain expansion-based methods [13], and we do compare to PackNet [12], which can be seen as very similar to expansion-based methods.
>
> ***2. Replay Baseline: Replay based on generative models [5-6] has not been considered as a baseline.***
>
> In this paper we look at online continual learning where you only have access to the data stream batch by batch (not task by task) which means that generative replay methods usually cannot be considered. This is why in previous work on online continual learning they are not compared against [7-11]. For example, Deep Generative Replay [6], learns a GAN at the end of each task and trying to learn one at the end of each batch would be very costly and probably will not perform well due the small amount of new data. Furthermore, in the experiments of the Deep Generative replay paper, their method is worse or comparable to experience replay in performance, which is a method we do compare against. Therefore, while generative replay methods are an important research direction they cannot be applied to what we look at in this paper.
>
> ***3. The writing and the presentation in the paper can be improved significantly. In terms of presentation, section 4 seems like an over-generalization of the methodology presented in section 5. Since the paper only discusses a simple variant of the general framework, which uses a class conditioned Gaussian to define the probabilistic model, section 4 seems a bit redundant.***
>
> We see this as a matter of taste; we prefer to state the method in its most general form as that helps to explain and show its key components and allows readers to see how else it could be applied/used. Therefore, we included a section to explain the general parts of our method, akin to previous work [14] before being explicit about the specifics of the DeepCCG method we explore in detail in the paper.
>
> ***4. Some of the claims over-state the contribution, for e.g. Pg 5 section 4 (last para): "our general approach can be used in task-agnostic settings as well" -- The task-agnostic continual learning is a much more challenging setting than task incremental. Without any empirical evidence, this claim is not well supported.***
>
> Thank you for pointing this out, we what we meant by this comment was to mention that it is possible to use our general approach to create a method for the task-agnostic setting, which, as we state in the paper, we see as a direction for future work. We did not mean to suggest that this new method would perform well, just that it is an interesting future direction of research. However, we understand this could be misleading so have edited the paper.
>
> ***5. The sample selection strategy based on the KL divergence of the two posteriors in equation 11 assumes that the training data in each task is same across the tasks? From equation 11, it seems that given a new task with large number of samples, the new memory would be biased towards the new task, which could easily cause catastrophic forgetting of previous tasks. However, the paper only assumed settings with equal number of samples across all tasks.***
>
> Our sample selection mechanism does not assume balanced training data across tasks and in-fact we look at the online setting where a method at a given point in time will often have seen an imbalanced amount of data from previous tasks and the current task. Additionally, the sample selection mechanism balances the number of samples between classes stored (as stated in Section 5, 1st para of sample selection, line 7) therefore, we think our method would be somewhat robust to different sized training sets for each task and/or class but have not explored this direction and leave it to future work as mentioned at the end of the sample selection paragraph in Section 6.3.

---

> > ### Author Response · Authors · 2022-11-15
> > **Author Response, Part 2**
> >
> > ***6. The paper only considers the task-incremental setting, which is known to be a relatively easier setup than other challenging continual learning settings, including class-incremental and task-free continual learning.***
> >
> > We believe that while the task-incremental setting might be seen as easier than class-incremental learning, it is still an important setting which many works look at [13] and as shown by our shifting window experiments, continual learning methods still have a way to go before fully solving the task-incremental setting.
> >
> > ***7. The notation in the paper can be improved: In equation 1, the marginalized likelihood is a function of the embedding network. Making this precise in the notation (e.g. $p_\phi(y|x,t_{(x,y)})$) would make the discussion that follows in section 4 clearer.***
> >
> > Thank you for this suggestion, we have updated the paper to include it.
> >
> > ***8. In section 6.1 under benchmarks: "in shifting windows ... there is no overlap between any two of the task datasets" -- Is this a typo? There is overlap with shifting windows, right?***
> >
> > Sorry, we realise now that this could be confusing, what we meant here is that the datasets for each tasks share none of the same examples. However, we understand using "overlap" here is confusing and so have updated the paper to change this.
> >
> > ***9. Novelty: The novelty seems limited given that the probabilistic model is based on a simplified class-conditional Gaussian model (LDA) which was also used in Ostapenko et al. (2020). Moreover, the idea of using a subset of samples to determine the conditional marginal log-likelihood (Equation 3) was also explored in Lofti et al., (2022), although its application in continual learning seems new.***
> >
> > While class conditional Gaussian models are nothing new, to the best of our knowledge we are the first to propose how to learn them probabilistically in a continual learning setting where the embedding function has to be learnt, which is a much tougher problem then when the embedding function is fixed as in Ostapenko et al. (2020). Also, as you have noted, while using the conditional marginal log-likelihood has been used in previous work, it has not been used in continual learning and has a special meaning in continual learning as it can be seen as a form of replay, which we show in the paper. Additionally, we propose a new sample selection mechanism, which we show in our ablations is necessary for DeepCCG to perform well. Therefore, we see the contributions of how to learn both the embedding function and class conditional Gaussian model together and the sample selection mechanism as novel, performative and of use to the continual learning community. Lastly, we also look at the shifting window setting which to the best of our knowledge has not been looked at before and show that currently continual learning methods perform badly on it compared to the multi-task upperbound, showing there is still problems to be solved in the online task-incremental setting.
> >
> > *** ***
> > We hope that, when you reconsider your review, that this answers your questions sufficiently, especially about the novelty of the method, and clarifies why we believe this paper is deserving of presentation at ICLR.
> >
> > [1] A Neural Dirichlet Process Mixture Model for Task-Free Continual Learning, (Lee et al., (2020)) [2] Continual Learning using a Bayesian Nonparametric Dictionary of Weight Factors (Mehta et al., (2021)) [3] Bayesian structure adaptation for continual learning, (Kumar et al. (2020)) [4] Hierarchical Indian Buffet Neural Networks for Bayesian Continual Learning, (Kessler et al, (2020)) [5] Efficient Feature Transformations for Discriminative and Generative Continual Learning, (Kumar et al., (2021)) [6] Deep Generative Replay, (Shin et al. (2017)) [7] Online Continual Learning with Maximal Interfered Retrieval (Aljundi et al., 2019a) [8] Gradient Based Sample Selection for Online Continual Learning (Aljundi et al,. 2019b) [9] On Tiny Episodic Memories in Continual Learning (Chaudhry et al., 2019a) [10] Online continual learning via candidates voting (He \& Zhu, 2022) [11] Efficient Lifelong Learning with A-GEM (Chaudhry et al., 2019b) [12] PackNet: Adding Multiple Tasks to a Single Network by Iterative Pruning (Mallya \& Lazebnik, 2018) [13] Continual learning: A comparative study on how to defy forgetting in classification tasks (Delange et al., 2021) [14] Variational Continual Learning (Nguyen et, al., 2018)

---

> > > ### Comment · Reviewer_rHsD · 2022-11-25
> > > **Thank you for the response.**
> > >
> > > Thanks for the response! I have read the rebuttal and other reviewers' comments. In general, I am satisfied with the clarification related to the sample selection strategy and data imbalance. *I am increasing the score to 5.*
> > >
> > > I have not increased the score further due to the following reasons:
> > > - The paper only considers the task-incremental setting, which is not applicable to the general setting of continual learning. Most recent works consider more general settings like class-incremental and task-agnostic continual learning.
> > > - I am not fully convinced with the justification for missing related work (expansion-based methods) and the missing replay baselines. These methods can be extended to learning when one only has access to the data stream in a task-incremental setting. Since the focus of this work is on task incremental setting, I think these baselines are highly relevant. (Also, the prior works [7,8,10] do not compare against expansion-based baselines because they do not consider task-incremental setup.) In any case, if these baselines can not be adapted to the setting considered in this work, I think the limitations of existing works should be discussed in the related work.
> > > - As also pointed out by the *reviewer juyJ*, the novelty in the methodology is limited.

---

> > > > ### Author Response · Authors · 2022-12-12
> > > > **Author Response**
> > > >
> > > > Thank you for reconsidering your review and for your additional comments, which we provide answers to below.
> > > >
> > > > ****10. The paper only considers the task-incremental setting, which is not applicable to the general setting of continual learning. Most recent works consider more general settings like class-incremental and task-agnostic continual learning.****
> > > >
> > > > We believe that the task-incremental setting is an important setting for continual learning [13] and show in our shifting window experiments that methods still do not perform well on it when tasks have more overlap, a challenging new research direction. Also, we look at online continual learning, where the learner receives data batch by batch, which is a harder and more general setting than the offline setting looked at in many other works. Additionally, recent work has shown that it is possible to convert task-incremental learning algorithms into class-incremental variants [15], by predicting the task using an OOD detector, getting better performance than standard class-incremental algorithms. This shows that there is not to much difference between the two settings in applicability or difficulty and that work on task-incremental learning can be transferred to class-incremental learning.
> > > >
> > > > ****11. I am not fully convinced with the justification for missing related work (expansion-based methods) and the missing replay baselines. These methods can be extended to learning when one only has access to the data stream in a task-incremental setting. Since the focus of this work is on task incremental setting, I think these baselines are highly relevant. (Also, the prior works [7,8,10] do not compare against expansion-based baselines because they do not consider task-incremental setup.) In any case, if these baselines can not be adapted to the setting considered in this work, I think the limitations of existing works should be discussed in the related work.****
> > > >
> > > > As mentioned before, we do not compare to expansion-based methods due to our assumption on fixed resource constraints and so focus on comparing and relating to methods which satisfy these constraints. Also, it is not clear to us that the reason why [7,8,10] do not compare against expansion based methods is that they look at the class-incremental setting as there has been work on using expansion based methods for class-incremental problems [16] and [7] state that they assume a fixed sized model, like our work, which is probably why they do not compare to expansion-based methods. However, we realise we could make this clearer in the paper and so will update it and we thank you for bringing this to our attention. Additionally, we do compare to PackNet, a competitive parameter-isolation method, which is closely related to expansion based methods and achieved the best performance in the study by De Lange et, al. [13] for parameter-isolation methods.
> > > >
> > > > ****12. As also pointed out by the reviewer juyJ, the novelty in the methodology is limited.****
> > > >
> > > > We believe that our paper is novel, in the context of being worthy of being published at ICLR. While we have already given a detailed reason why in our answer to Question 9, we reiterate that we propose a new loss using a conditional log marginal likelihood term which to the best of our knowledge has not been used before in continual learning. Secondly, we propose a new sample selection method which to the best of our knowledge has not been used before. Finally, we propose a new setting, the shifting window setting which shows why there is still work to be done in online task-incremental settings when there is increased task overlap.
> > > >
> > > > *********
> > > >
> > > > We hope that we have addressed your concerns more fully now, especially regarding to looking at the online task-incremental setting and the novelty of our paper and that these additional comments are taken into account when giving your final review.
> > > >
> > > > [15] A Theoretical Study on Solving Continual Learning (Kim et, al, 2022) [16] Expert Gate: Lifelong Learning with a Network of Experts (Aljundi et, al. 2017)

---

### Official Review · Reviewer_U88A · 2022-12-08

**Confidence:** 5
**Correctness:** 3
**Technical Novelty And Significance:** 2
**Empirical Novelty And Significance:** 2
**Recommendation:** 5

**Clarity, Quality, Novelty And Reproducibility:**

The paper has clear writing, novelty is limited and proper baseline of close work and recent works are missing.
The code is provided hence reproducibility may not be an issue.

**Strength And Weaknesses:**

Pors:

1: The model is simple and requires a small memory buffer to overcome the catastrophic forgetting.

2: Easy to follow and ablation for the sample selection strategy w.r.t. reservoir shows the improvement. Sample selection method is intuitive and it may be useful in the other replay based approach. However, practical time complexity may be a bottleneck.

3: For the reproducibility, they provided the code.

Cons:

1: The paper handles the easiest setting of the continual learning (CL) where task id are provided during inference, which is not practically useful. In this setting, a simple expansion based approach [a,b,c,d] without any replay, model achieves the performance near the upper-bound without using the pretrained backbone. In this paper, none of these methods are compared. Replay is required for the comparatively much complex setting, like class incremental learning or task-free continual learning.

2: The model is not end-to-end, it learns the embedder and classifier separately. The embedder learns the class specific cluster representation and class conditional Gaussian learns the prototype classifier. Also, learning only the mean is too simple and it may not handle the complex scenarios.

3: The sample selection strategy requires finding the subset of samples to maximize the posterior. Is there any optimization based model to select this subset of samples? If not, it’s like a brute-force model and sample selection is too costly.

4: Novelty is limited conditional marginal likelihood was explored in earlier work, sample selection approach seems novel but the complexity of it may be a bottleneck.

5: The baselines are weak and no recent model are included. How the author created the baseline for the previous work and upper-bound? The implementation details of the baseline are missing, do the author did the proper hyperparameter search for the baseline?

[a] Calibrating CNNs for Lifelong Learning, NeurIPS-20

[b] Ternary Feature Masks: zero-forgetting for task-incremental learning, CVPRW-21

[c] Supermasks in Superposition, NeurIPS-20

[d] Continual Learning using a Bayesian Nonparametric Dictionary of Weight Factors, AISTATS-21



**Summary Of The Paper:**

The paper proposes a deep class conditional model for the online task's incremental learning. The primary motivation of the paper is to learn the feature embedder since a fixed feature extractor is not useful in the dynamic environment. In this work, the model leverages the replay buffer to overcome the catastrophic forgetting and the buffer samples are selected by maximizing the posterior between the subset of samples and task samples. They used jointly the buffer and batch samples to update the embedder network parameter. Finally, the samples are classified by learning the prototype/mean of the class.

**Summary Of The Review:**

The paper uses the simplest setting of CL and without replay recent model shows the result close to the upper-bound.

Recent comparison with the similar and recent work is missing

Paper has some novelty, but that is not significant for the publication.

---

> ### Author Response · Authors · 2022-12-12
> **Author Response, Part 1**
>
> Dear reviewer, thank you for your comments and questions which we think will improve the quality of our submission. We provide answers to your comments and questions below.
>
> ****1. The paper handles the easiest setting of the continual learning (CL) where task id are provided during inference, which is not practically useful. In this setting, a simple expansion based approach [a,b,c,d] without any replay, model achieves the performance near the upper-bound without using the pretrained backbone. In this paper, none of these methods are compared. Replay is required for the comparatively much complex setting, like class incremental learning or task-free continual learning****
>
> We think the online task-incremental setting we look at is interesting and useful. One reason for this is that we look at the online variant of the setting where data is received batch by batch. We see the online setting as more realistic, harder than the offline setting and under-explored as many offline approaches, like many expansion-based methods, cannot be used due to assuming that they can see and iterate over the all of the data of a task at the same time. Additionally, as shown by recent work [e], it is possible to convert task-incremental methods to be class-incremental, by using an OOD detector to identify tasks, getting SOTA performance and showing that work in task-incremental learning can be directly transferred to class-incremental learning. Hence, the difference in difficulty between the two settings might be overstated [e]. Lastly, in the experiments on our new shifting window setting, we show that task-incremental learning methods still do not perform well when there is a greater overlap between tasks, especially for the PackNet parameter-isolation method [h], showing there is more work to be done in this setting.
>
> The reason we do not compare to expansion-based methods is that we assume fixed resource constraints. Therefore, given the large amount of work on continual learning we focus on relating and comparing to work which have the same assumptions, like many other works on replay methods [f, g, i]. Additionally, we do compare to a competitive parameter-isolation method PackNet [h], which is closely related to expansion based methods and achieved the best performance for parameter-isolation methods in the study by De Lange et, al. [i].
>
> ****2. The model is not end-to-end, it learns the embedder and classifier separately. The embedder learns the class specific cluster representation and class conditional Gaussian learns the prototype classifier. Also, learning only the mean is too simple and it may not handle the complex scenarios.****
>
> While we do not think it matters too much if a method is learnt end-to-end or not, our method is learnt end-to-end. This is because to update the embedding function we propagate gradients back through the class conditional Gaussian classifier (i.e., the predictive posterior of our model, given $M_j/R_j$). Secondly, because we learn the embedding function, only learning the means of the Gaussians is sufficiently flexible as the embedding function can be seen as implicitly learning a covariance and you can increase the complexity of the model by increasing the size of the neural network embedding function.
>
> ****3. The sample selection strategy requires finding the subset of samples to maximize the posterior. Is there any optimization based model to select this subset of samples? If not, it’s like a brute-force model and sample selection is too costly.****
>
> Yes, we agree with you that brute force search for the best subset would be too costly. Therefore, as explained in the second part of Section 5 of the paper, we relax the problem by using a lasso loss (Equation 12) which can be optimised efficiently and is shown in our ablations to perform well.
>
> ****4. Novelty is limited conditional marginal likelihood was explored in earlier work, sample selection approach seems novel but the complexity of it may be a bottleneck.****
>
> We are unaware of any other continual learning method which uses our conditional marginal log-likelihood loss hence we believe our train procedure for DeepCCG is novel. Additionally, we also propose a new setting for continual learning, the shifting window setting which shows that in task incremental learning there is still work to be done when tasks have more overlap. Therefore, we believe our paper is sufficiently novel as the training of DeepCCG, the shifting window setting and, as you state, the sample selection mechanism are novel contributions and we believe are of interest to the continual learning community.

---

> > ### Author Response · Authors · 2022-12-12
> > **Author Response, Part 2**
> >
> > ****5. The baselines are weak and no recent model are included.****
> >
> > To the best of our knowledge, for the online task-incremental continual learning setting ER-reservoir is highly competitive and is still often SOTA, which is why our paper has it as our main comparison. Additionally, we compare to GSS, which is the best sample selection method we know of for online task-incremental settings. Lastly, we note that currently a lot of focus in continual learning is on offline, class-incremental and/or other types of settings, which is perhaps why there is less by way of more recent work that is consistently SOTA for the setting we consider.
> >
> > ****6. How the author created the baseline for the previous work and upper-bound? The implementation details of the baseline are missing, do the author did the proper hyperparameter search for the baseline?****
> >
> > We performed grid searches for the hyperparameters, which is described in Appendix C, and we re-implemented all the baselines so that they could be tested on our new shifting window setting. Other experimental details are described in Section 6.1.
> >
> > ********
> >
> > We hope that we have sufficiently answered your questions, especially about the computational complexity of our sample selection mechanism, the use of the online task-incremental setting and the novelty of our method, and that our answers are taken into account when reconsidering your review.
> >
> >
> > [a] Calibrating CNNs for Lifelong Learning, NeurIPS-20
> > [b] Ternary Feature Masks: zero-forgetting for task-incremental learning, CVPRW-21
> > [c] Supermasks in Superposition, NeurIPS-20
> > [d] Continual Learning using a Bayesian Nonparametric Dictionary of Weight Factors, AISTATS-21
> > [e] A Theoretical Study on Solving Continual Learning (Kim et, al, 2022)
> > [f] Online Continual Learning with Maximal Interfered Retrieval (Aljundi et al., 2019a)
> > [g] On Tiny Episodic Memories in Continual Learning (Chaudhry et al., 2019a)
> > [h] PackNet: Adding Multiple Tasks to a Single Network by Iterative Pruning (Mallya \& Lazebnik, 2018)
> > [i] Continual learning: A comparative study on how to defy forgetting in classification tasks (Delange et al., 2021)

---

> > ### Comment · Reviewer_U88A · 2022-12-13
> > **Re: Part:1**
> >
> > Comparision:
> >
> > The expansion and replay based approach both uses extra memory, one uses it for the parameter storage and another for the previous sample storage. If the order of extra parameter are similar, then I think there should be a comparison. The paper is using the extra data (replay), however; the expansion based model uses extra parameter. Both are using extra information (parameter/data) for the to mitigate the forgetting. I believe that comparison with the expansion based model is fair.
> >
> > The expansion based model with 4% extra parameter shows much better result, however model used 10% data storage.
> >
> > If the author has problem to compare the model with expansion based approach, please compare the model with [a,b,c ], these are the replay based model.
> >
> > [a] Learning Fast, Learning Slow: A General Continual Learning Method based on Complementary Learning System, ICLR22
> > [b] Dark Experience for General Continual Learning: a Strong, Simple Baseline, NeurIPS20
> > [c] Gdumb: A simple approach that questions our progress in continual learning. ECCV20
> >
> > Novelty:
> >
> > The proposal of a shifting window is not a significant contribution, I believe that is trivial.

---

> > > ### Author Response · Authors · 2022-12-13
> > > **Author Response**
> > >
> > > Thank you for these extra comments.
> > >
> > > ****7. The expansion and replay based approach both uses extra memory, one uses it for the parameter storage and another for the previous sample storage. If the order of extra parameter are similar, then I think there should be a comparison. The paper is using the extra data (replay), however; the expansion based model uses extra parameter. Both are using extra information (parameter/data) for the to mitigate the forgetting. I believe that comparison with the expansion based model is fair. The expansion based model with 4% extra parameter shows much better result, however model used 10% data storage.****
> > >
> > > Yes, we agree with you that one of the main differences between expansion-based and replay methods is what additional data they store. We follow previous work on replay methods [f, g, i] which do not compare to expansion-based methods and assume a fixed model and fixed resource constraints; therefore, meaning expansion-based methods cannot be used. Additionally, we want to emphasise that we do compare to PackNet a competitive parameter-isolation method in all our experiments.
> > >
> > > ****8. If the author has problem to compare the model with expansion based approach, please compare the model with [a,b,c ], these are the replay based model.****
> > >
> > > Thank you for suggesting these extra comparisons, which are all interesting pieces of work, we would compare to them but as the reviewing stage has finished, we do not have time to run experiments now. Additionally, we would like to say that Zhang et, al. [k] show DER does not perform that well in the online setting, performing quite a bit worse than ER and that Gdumb might require a larger memory than we use to perform well, in their experiments they store twice as many examples.
> > >
> > > ****9. Novelty: The proposal of a shifting window is not a significant contribution, I believe that is trivial.****
> > >
> > > We agree with you that the shifting window is quite a simple new setting. However, the reason we believe it is of use to the continual learning community is, as stated in our previous answers, that our experiments on it show that current continual learning methods perform much worse in this setting than the standard disjoint tasks setting, in comparison to the upper bound. We believe this is an interesting new finding which shows that when there is more task overlap current methods perform worse, which we see as a novel contribution of the paper. Additionally, we hope that in our answer to your first question about novelty (Question 4) we fully answered your question about the novelty of DeepCCG and the new continual learning loss used in particular.
> > >
> > > [k] A simple but strong baseline for online continual learning: Repeated Augmented Rehearsal (Zhang et, al. 2022)

---

> > ### Comment · Reviewer_U88A · 2022-12-13
> > **Re: Part1**
> >
> > Task Incremental and Class Incremental:
> >
> > it is just easy to say:
> >
> >
> > "It is possible to convert task-incremental methods to be class-incremental, by using an OOD detector to identify tasks, getting SOTA performance and showing that work in task-incremental learning can be directly transferred to class-incremental learning."
> >
> > When we try the OOD, we face a lot of problems. We know how well the neural network model behaves for the OOD detection. I request to author instead of claim if they show the result that would be much better.

---

> > > ### Author Response · Authors · 2022-12-13
> > > **Author Response**
> > >
> > > Thank you for this comment on our answer to Question 1, we hope the rest of the answer was satisfactory. In context to your comment above, as stated in our answer to Question 1, Kim et, al [e] have already shown that this method of converting a task-incremental method to a class-incremental variant works, in their experiments getting a large performance improvement over other class-incremental methods and their paper provides theoretical justification as well. Therefore, there is already empirical evidence that shows this method works but we agree with you that it would be nice to show results when using it on DeepCCG and we are actively pursuing this direction.

---

> ### Author Response · Authors · 2022-12-13
> **Author Response, to extra comments**
>
> Thank you for your additional comments and questions, which we have replied to separately, we just wanted to say here that while the reviewing period has now ended, we hope that we have sufficiently answered your extra questions and they are considered when giving your final review.

---

### Decision · Program_Chairs · 2023-01-20

**Decision:**

Reject

**Justification For Why Not Higher Score:**

The paper does have the makings of a nice paper but, due to the reasons given in the meta-review, the paper falls short of the acceptance criterion.

**Justification For Why Not Lower Score:**

N/A

**Metareview: Summary, Strengths And Weaknesses:**

This paper presents a method for online continual learning for the setting when the task ids are known (the task-incremental setting). The basic idea is to model each class using a class-conditional Gaussian in an embedding space which is learned in an online fashion (instead of using a pre-trained embedding function).

The paper received detailed reviews from reviewers who are experts in continual learning. While the reviewers appreciate the idea, there were also several concerns, one of which being the fact that the paper considers the task-incremental setting which is considered much more easy (and less practical) than the class incremental/task agnostic setting. In fact, there is a lot of recent interest and recent work on online class-incremental learning. Although the authors in the response say that their method can be extended to this setting by having a task-id predictor, it needs to be seen how such an approach would perform.

The reviewers also had some concerns about missing baselines. The authors responded to these concerns but the reviewers remain unsatisfied from the response.

In the end, after considering the reviews, the author response, and the discussions, it was felt that the paper, despite its interesting ideas, still falls short of the acceptance criterion. Also, none of the reviewers championed the paper or argue for acceptance.

I think the paper does have a nice, principled Bayesian framework; however, the above-mentioned issues also weaken it. Overall, it has makings of a good paper but is not there yet. The paper would be stronger if the proposed method can be adapted to work for the (online) class-incremental learning setting as well, in addition to empirically showing that it is competitive to recently proposed online class-incremental learning methods. The authors are encouraged to explore this possibility to improve their method and submit at another venue.

Although, unfortunately, the paper fell below the acceptance bar, I also deeply appreciate the authors for engaging in a constructive and open discussion with the reviewers.